# Facial soft tissue thickness in forensic facial reconstruction: Impact of regional differences in Brazil

Deisy Satie Moritsugui[1][◉]*, Flavia Vanessa Greb Fugiwara[1‡], Flávia Nicolle Stefani Vassallo[1‡], Luiz Eugênio Nigro Mazzilli[1◉], Thiago Leite Beaini[2◉], Rodolfo Francisco Haltenhoff Melani[1◉]

1 Laboratory of Forensic Anthropology and Odontology (OFLAB), Department of Social Dentistry, School of Dentistry, University of São Paulo, São Paulo, São Paulo, Brazil, 2 Department of Preventive and Social Dentistry, School of Dentistry, Federal University of Uberlândia, Uberlândia, Minas Gerais, Brazil

◉ These authors contributed equally to this work.
‡ FVGF and FNSV also contributed equally to this work.
* dsmoritsugui@usp.br

**Data Availability Statement:** The minimal data set are within the manuscript file and in Supporting Information files.

## Abstract

Forensic facial reconstruction aims to assemble and provide the appearance of a face over a skull, in order to lead to recognition of that individual, making possible the application of primary identification methods. The scientific literature presents facial soft tissue thickness (FSTT) tables for reference from a range of different geographic regions. However, the consensus on its importance or on how to use specific population data related to FSTT is not unanimous. Brazil is formed by geographic regions with diverse populations, which are reflected in facial features. This paper aimed to measure and compare FSTT of distinct Brazilian samples to ascertain the need for specific data sets for different regions. A specific protocol for cone beam computed tomography was used to standardize measurement, and it was applied in a sample of 101 subjects. The FSTT measurements of a Brazilian population from the Midwest Region was compared to a previous sample from Southeast, which was collected using the same protocol. High compatibility was observed when comparing the averages of FSTT among samples of these two different geographic regions. Regarding age groups, notable differences on the medium and inferior face were observed in females. Minor variances found are unlikely to affect the practice of forensic facial reconstruction. Facial features, such as eyes, lips, nose, and skin may also be relevant in the differentiation of people from these two areas in Brazil. Therefore, concerning the Southeast and Midwest Brazilian regions, the need to apply different data sets is unnecessary.

## Introduction

The forensic facial reconstruction (FFR) has an important role in recognizing individuals who are unable to be identified by other methods. Advanced postmortem decomposition process or the lack of antemortem information are its main indication for forensic purposes [1, 2]. The

**Funding:** The author(s) received no specific funding for this work.

**Competing interests:** The authors have declared that no competing interests exist.

technique is based on the assembling of an approximated soft tissue topography over the unknown skull, with the intent of reconstructing its characteristics to be as similar as possible to the individual at the time of their death. Once the reconstruction is finished, a public campaign is issued, aiming to illicit a response from the public as to whom the remains might belong. Following this, a formal identification can then take place [1, 3, 4].

An approximated face is achievable by using various techniques. The 2D representation is based on a picture of the skull, whereas the 3D representation can be divided into manual or digital methods. The manual method involves direct modelling and sculpture, applying clay or Plasticine over the skull or a model of the skull [1]. More recently, a computerized face sculpture can be generated by using specialized modeling software, and different user interfaces can adapt the procedures to resemble manual modeling. Tablets and the assistance of haptic devices may be used in the 3D context to add soft tissues to a model of the skull [5]. However, independently of the method used, it is imperative to establish parameters and guidance such as the facial soft tissue thickness (FSTT), that serves as reference over specific craniometric landmarks [6], Those FSTT measurements are usually populational-based averages, but from this general information it is often expected to deliver a precise and reliable assistance during FFR [7].

FSTT charts in the scientific literature cover many craniometric landmarks, allowing it to be organized by sex, age, ancestry, BMI, among other facial features considered to be significant in an FFR. Many aspects of the method that seem to be related to the success rate or to the probability of recognition of the reconstructed faces. Studies have indicated that FSTT data are particularly influenced by the population groups under study [8–12], but the debate on the subject has far from achieved a consensus. Therefore, the expert should consider using of FSTT based on the population to which the body can be correlated by previous anthropological exams because specific FSTT charts could produce better results. However, in those studies, it is remarkable that many describe different methodologies, including the original measurement protocols, the source and type of imaging exam and the relation of regional charts, and the success of the FFR has not yet been established. Some researchers do not believe there is enough evidence to justify the use of different population data [13], but it is unclear how this might apply in countries like Brazil, with diverse and complex demographics.

The Brazilian territory has continental proportions, and it is divided into geographical regions (north, northeast, southeast, south, and Midwest), each one with its diverse environments and populations.

To understand the local context better, it is important to remember that the original population was based on several Indigenous Brazilian with different anthropological features among the groups found in the territory. The European colonization favored the concept of a mixed ancestry population, mostly influenced by the presence of Portuguese, Dutch and French. After the world wars of the 20th century, the presence of immigrants from Japan, Eastern Europe, the Middle East, Italy and Germany became important [14]. Their distribution was not even among the regions and, therefore, it is clear that each segment of Brazil has its own physical features that represent the dominant population groups, with different characteristics [15]. The increasing and continuous lack of ancestral uniformity, as it is observed in Brazil, raises the question whether or not the use of a single STT chart is appropriate for the estimation of facial morphology [16].

This study considers that the craniofacial morphology is determined by hereditary features and influenced by the environment [11], affecting the skull and the face. The combination of these anatomical differences makes the individual unique, leading to a wide variation of facial profiles across the country [17]. This justifies deeper investigation and the collection of data from different regions. With the intent of verifying the need for various FSTT charts on

reconstruction from each geographic region, this research aimed to measure the FSTT in a Brazilian sample from the Midwest region and to compare to a previous sample from the Southeast region [18] using the same cone-beam computed tomography (CBCT) measuring protocol.

## Materials and methods

The sample was composed of 101 CBCT exams of Brazilian subjects from the Midwest region, available in a previously established anonymous data bank in Digital Imaging Communication in Medicine (DICOM) format. Table 1 shows sample distribution.

The exams were acquired from records of clinical diagnosis, and, therefore, no participant was exposed to ionizing radiation for this research. The image protocol had a 23 cm x 17 cm FOV field of view, allowing the visualization of all included anatomical structures and landmarks from the supraglabellare to menton in the vertical axis and posterior to the supraglenoid structure. The images were obtained with an i-CAT Next Generation tomograph with 17.8 second exposition and voxel of 0.3 mm, and they were exported on Digital Imaging and Communication in Medicine (DICOM) format. CBCT scans allow image acquisitions with the individual in a sitting or standing position. Despite containing stabilizing devices for the head, such as the cephalostat and the chin support, these were not used in our sample, in order to avoid the effects of compression of facial soft tissues.

The research was approved by the Committee of Ethics in Research of the School of Dentistry of University of São Paulo (CAAE 33433320.8.0000.007 and 33433320.8.0000.0075). The inclusion criteria were established by observing the facial integrity of the research subjects. Therefore, this project included the exams of adult patients (18 and older) with required presence of certain structures: the central incisors and second molars (including both natural and prosthetically rehabilitated elements), no visible facial syndromes (hard or soft tissue), with no severe asymmetry and that had not been submitted to orthognathic surgery. Furthermore, the research included exams of both male and female patients in balanced groups. Thus, the exclusion criteria rejected exams that did not fit all the inclusion criteria or had quality issues that could prejudice the measuring of soft tissues.

The selection and measurements were performed with the assistance of the Horos software (version 3.3.6, 64 bit, Horos Project, Purview, Annapolis, USA) installed on an iMac computer (macOS Catalina–version 10.25.2, a 3,4 GHz Intel Core i7 Quad-Core processor, 16 GB RAM) and the 3D reconstruction tool 3D volume rendering through the skin and bone filters. Horos software is an open access version of OsiriX (DICOM viewer software) and has demonstrated to be a good alternative for craniofacial anthropological analyses, both in hard and soft tissues [19].

The exams were divided by sex into three age groups (from 18 to 30 years old, from 31 to 40 years old, 41 and older) following the original research group divisions [18], whose control sample, from Southeast Brazilian region, was composed of 100 exams equally distributed among the groups.

**Table 1. Sample distribution by sex and age group.**

| Sex | Male | Female | Total |
|---|---|---|---|
| **Age group** | | | |
| 18 to 30 years | 18 | 22 | 40 |
| 31 to 40 years | 13 | 14 | 27 |
| 41 and older | 14 | 20 | 34 |
| **Total** | 45 | 56 | 101 |

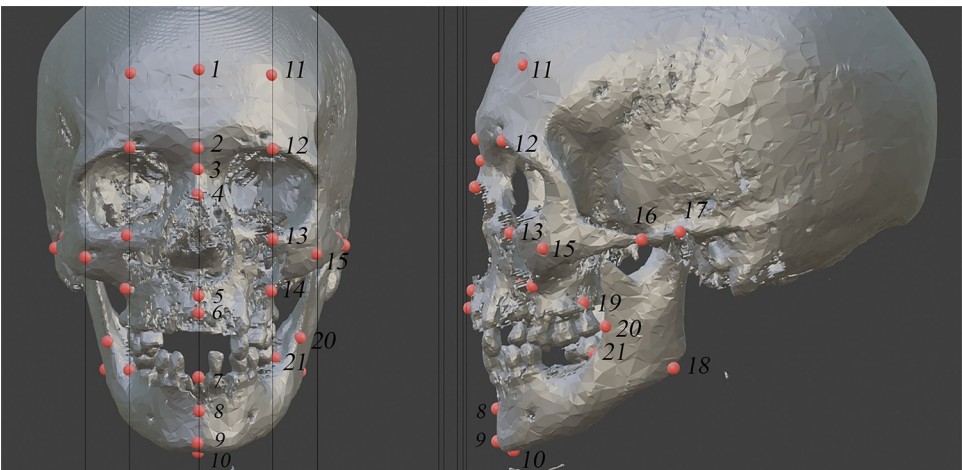

**Fig 1. Craniometric landmarks measured.**

The protocol established by Beaini et al. [18] was used to create a standardized measurements of FSTT in CBCT. In this guide, the skull images were positioned with the Frankfurt Horizontal Plane to avoid the influence of misplacement on multiplanar view. The measurements of FSTT were executed in the same craniometric landmarks studied by Rhine and Campbell [20]. Thirty-two craniometric landmarks were measured, ten located on the midline and eleven bilateral landmarks. Figs 1–3 present the craniometric landmarks and measurement direction, whereas Table 2 describes each landmark and the measurement directions according to the applied protocol.

The DICOM files from CBCT were imported into the Horos software, according to the protocol by Beaini et al. [18], a window level of 500 e window width of 3500 were applied to all exams to favor the visualization of both hard and soft tissue limits. The acquired volume was reconstructed on a 3D MPR window (multiplanar reconstruction) and visualized with the aid of a MIP tool (maximum intensity projection) to perform the volume repositioning to the Frankfurt Horizontal Plane, as previously mentioned.

An initial calibration was made in the form of inter- and intraobserver tests. The measurements for this research were collected only with a high concordance between observers. The evaluation and adjustment of inter and intra-examiners were achieved by testing the thirty-two landmarks in fifteen exams (15% of the sample) and testing the intraclass correlation coefficient (ICC) for each point. Regarding intra-examiner assessment, the main examiner measured the fifteen exams twice, considering a one week time interval between the measurements. The main examiner's previous training of the method was consistent with the original examiner.

The data were organized in a spreadsheet, using the software Microsoft Excel, and statistical analysis was performed with aid of the SPSS statistic pack, establishing a reliability level of 95%.

Using the Shapiro-Wilk test, the normality distribution sample was checked for every landmark, with $P < 0.05$, which implies alteration of normality. In the nonnormal distribution landmarks, the bootstrap method was used (1000 resampling– 95% CI BCa) to obtain reliable average values.

To search for sex related differences, the $t$-test was applied on landmarks with normal distribution, whereas the Mann-Whitney test was used with nonnormal distribution data. The variance analysis from a one-way ANOVA was applied to investigate any differences among

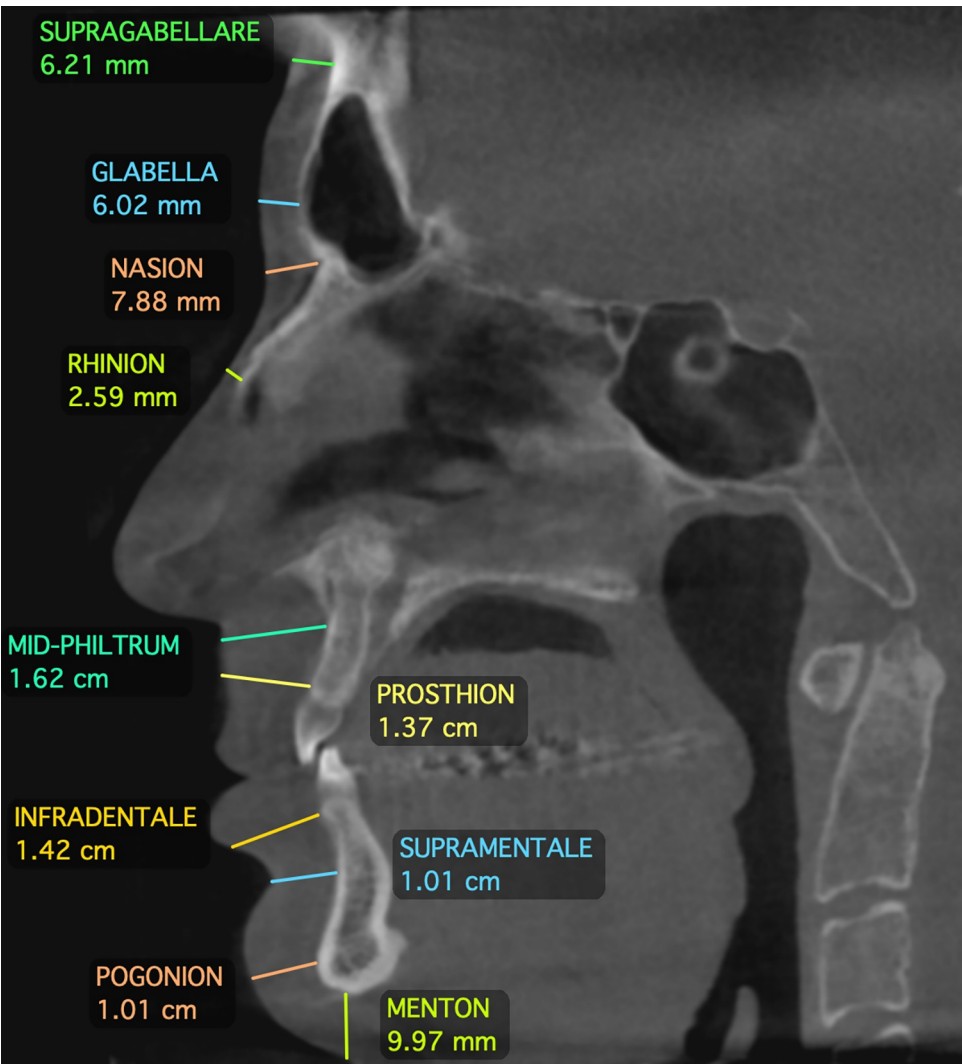

**Fig 2. Sagittal landmarks measurements.**

the average FSTT, by landmark, on the three age groups (18 to 30 years, 31 to 40 years and 41 and older).

Concerning the regional differences, the research data were compared to the raw data of the Southeast region [18] to which full access was granted. A *t*-test was applied on the comparison of FSTT averages among the Southeastern [18] and Midwestern (this study) regions, for both sexes.

## Results

The concordance of the measurements verified by the ICC according to each landmark measurement revealed intra- and interexaminer agreement above 95% (data in S1 Table), demonstrating reliability among the measurements. Fig 4 presents the individual ICC records.

For male subjects, the Shapiro-Wilk test revealed that, from the thirty-two analyzed landmarks, two presented nonnormal distribution (lateral orbital and ectomolare[2]). Female subjects presented nonnormal distribution at the landmarks rhinion, mid-infraorbital, zygion and

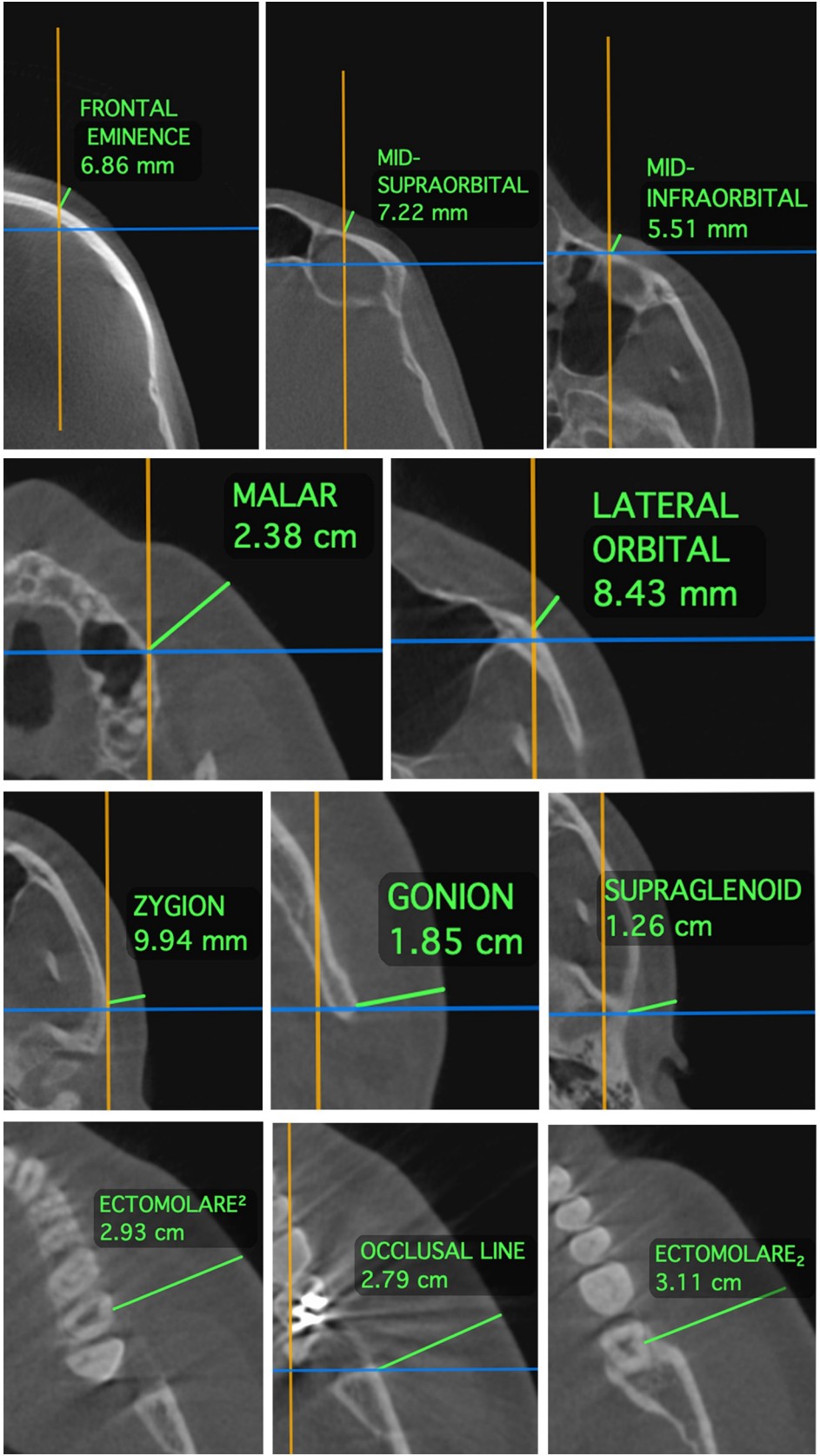

**Fig 3. Bilateral landmarks measurements.**

**Table 2. Craniometric landmarks, descriptions, and measurement direction.**

| | Landmark | Description | Measurement direction |
|---|---|---|---|
| | **Median points** | | |
| 1 | **Supraglabellare** | Deepest part of the supraglabella fossa in the median plane | Perpendicular to the bone surface |
| 2 | **Glabella** | Most projecting median point on lower edge of the frontal bone | Perpendicular to the bone surface |
| 3 | **Nasion** | Intersection of the nasofrontal sutures in the median plane | Nasal soft tissue fold |
| 4 | **Rhinion** | Most rostral point on the internasal suture | Perpendicular to the bone surface |
| 5 | **Mid-Philtrum** | Deepest point of the maxillary alveolar bone | Mid-Philtrum or subnasale |
| 6 | **Prosthion** | Median point between the central incisors on the anterior margin of the maxillary alveolar rim | Upper vermilion border |
| 7 | **Infradentale** | Median point at the superior tip of the septum between the mandibular central incisors | Lower vermilion border |
| 8 | **Supramentale** | Deepest median point in the groove superior to the mental eminence | Chin fold |
| 9 | **Pogonion** | Most anterior median point on the mental eminence of the mandible | Perpendicular to the bone surface |
| 10 | **Menton** | Most inferior median point of the mental symphysis | Vertical direction or to neck fold |
| | **Bilateral Points** | | |
| 11 | **Frontal Eminence** | Most projecting point of the frontal bone, at a line that vertically bisects the orbit | Perpendicular to the bone surface |
| 12 | **Mid-supraorbital** | Point on the anterior aspect of the superior orbital rim, at a line that vertically bisects the orbit | Orthoradial direction |
| 13 | **Mid-infraorbital** | Point on the anterior aspect of the inferior orbital rim, at a line that vertically bisects the orbit | Orthoradial direction |
| 14 | **Malar** | Intersection of the maxillary alveolar process and zygomatic process, at a line that vertically bisects the orbit | Perpendicular to the bone surface |
| 15 | **Lateral Orbital** | On the center of the zygomatic bone, at a line that vertically crosses the lateral orbital margin | Orthoradial direction |
| 16 | **Zygion** | The most lateral point on the zygomatic arch | Orthoradial direction |
| 17 | **Supraglenoid** | Most rostral point on the superior rim of the glenoid cavity | Orthoradial direction |
| 18 | **Gonion** | Point on the rounded margin of the angle of the mandible, bisecting two lines one following vertical margin of ramus and one following horizontal margin of the corpus of mandible | Perpendicular to the bone surface |
| 19 | **Ectomolare[2]** | Most lateral point on the buccal alveolar margin, at the center of the superior second molar position | Orthoradial direction |
| 20 | **Occlusal Line** | Most anterior point on the mandibular ramus, where the occlusal line meets de mandible | Orthoradial direction |
| 21 | **Ectomolare$_2$** | Most lateral point on the buccal alveolar margin, at the center of the inferior second molar position | Orthoradial direction |

gonion. In nonnormal distribution landmarks, the bootstrap method (1000 resamples– 95% CI BCa) was used to obtain reliable mean values. Such statistical procedure assigns consistent results, through the process of random resampling, because the process tends to correct normality deviations of the sample [21].

Table 3 exhibits the FSTT averages as the main result of this research. It has been divided by sex with SD, SE and *P* values, corresponding to each test, verifying statistically significant differences between the female and male averages, after bootstrapping procedure.

The FSTT analysis indicates the male sample exhibited higher FSTT values compared to the female sample, except for the orbital lateral, which demonstrated a difference among the averages of 0.8 mm. The major thickness discrepancies between the sexes were found on landmarks the gonion, ectomolare[2], ectomolare$_2$ and occlusal line, in which mean differences were above 3 mm.

Regarding age, an ANOVA was used, on both sexes, to compare averages between the age groups (18 to 30 years, 31 to 40 years and 41 and older). A direct relation between FSTT and ageing process can be seen, particularly among females. The mid-philtrum, prosthion an ectomolare[2] landmarks presented a decrease of FSTT as the age advanced.

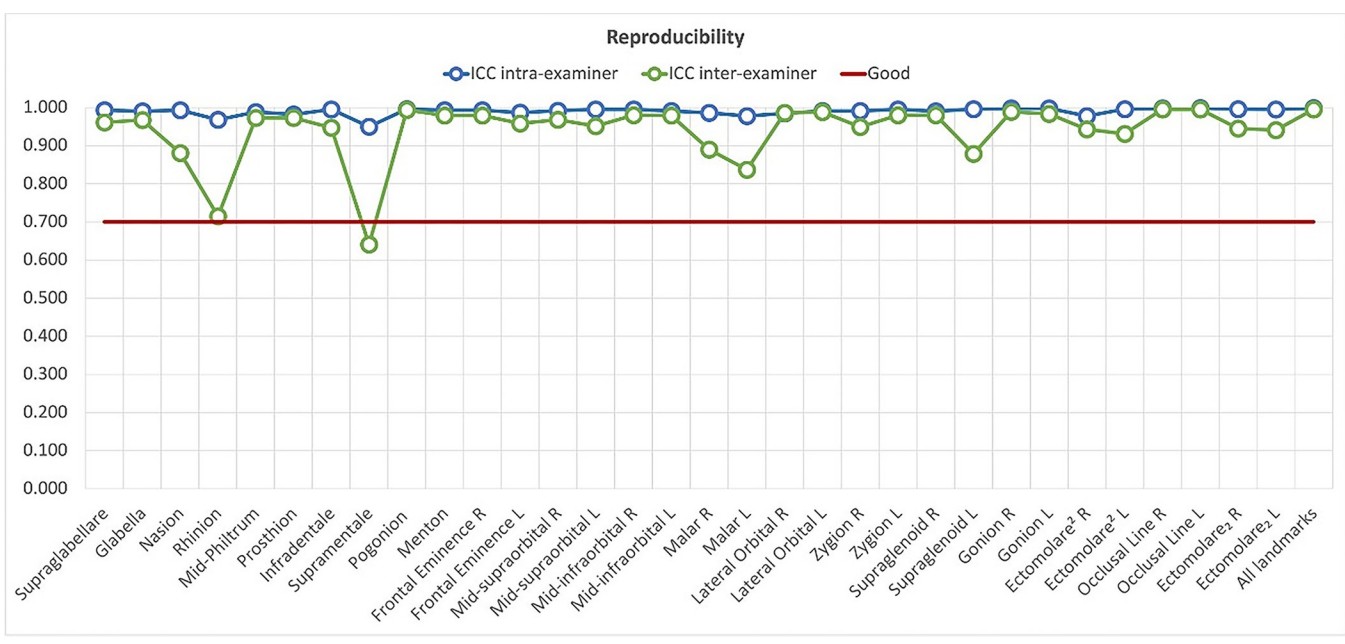

**Fig 4. Concordance level for each landmark in intra- and interobserver analysis.**

Figs 5 and 6 (data in S2 Table) show the comparison by sex among age groups. Quantitatively, a direct relation is noted between age and divergence of FSTT specifically for females, on mid-philtrum, prosthion and ectomolare[2] landmarks, which showed a decrease of FSTT as the age advanced.

Table 4 presents FSTT averages, on each landmark in comparison to the Southeastern sample (SE) [18] and the Midwestern (MW) (this paper), by sex, as well as $P$ values ($t$ test).

## Discussion

FSTT is one of the most studied subjects in the FFR field [22]. It provides quantitative data for the technique [13], but there are still some geographical regions that have not been studied. Studies have confirmed that FSTT measures obtained by using CBCT result in reliable and reproducible data [7, 18, 23, 24].

Several studies have verified the presence of FSTT variances among different populations, suggesting the importance of using specific values for each one [8, 10–12, 25, 26]. Some of these studies compared data of FSTT averages that were obtained with different measurement protocols with inevitable bias to those conclusions [13]. This may explain why there are other studies that contradict this hypothesis and do not support that specific FSTTs are needed for different populations for the FFR [27–29]. The FSTT measurement method is important to the analysis of the specificity among different studies [29]. Our study was able to compare regional FSTT using the same method from a previous study, eliminating the method variant.

The concordance results from the interexaminer ICC analysis showed the Beaini protocol [18] can provide reproducibility and that it is recommended for future research. The intra-examiner ICC analysis demonstrated high concordance on the measurements made by the main examiner, in accordance with other studies that utilized the CBCT with the same objective [18, 30]. Nevertheless, it was a result of practical experience gained on previous training, which may be necessary and well demonstrated in future studies.

**Table 3. Mean FSTT (millimeters and percentage), SD, and SE divided by sex.**

|  | Female | | | Male | | | Mean difference (mm) ** | Mean difference (%) *** | P value |
|---|---|---|---|---|---|---|---|---|---|
|  | Mean | SD | SE | Mean | SD | SE |  |  |  |
| Supraglabella | 4.5 | 0.97 | 0.14 | 5.5 | 1.14 | 0.23 | -1 | 18.1 | 0.00* T |
| Glabella | 4.9 | 0.77 | 0.1 | 5.9 | 0.95 | 0.14 | -1 | 16.9 | 0.00* T |
| Nasion | 6.2 | 0.97 | 0.13 | 8.1 | 1.24 | 0.18 | -1.9 | 23.4 | 0.00* T |
| Rhinion | 1.7 | 0.43 | 0.06 | 2.3 | 0.58 | 0.09 | -0.5 | 26.0 | 0.00* M |
| Mid-Philtrum | 13.1 | 1.82 | 0.24 | 15.7 | 2.22 | 0.33 | -2.7 | 16.5 | 0.00* T |
| Prosthion | 10 | 1.58 | 0.21 | 13.3 | 2 | 0.3 | -3.3 | 24.8 | 0.00* T |
| Infradentale | 9.8 | 1.34 | 0.18 | 11.9 | 1.5 | 0.22 | -2.1 | 17.6 | 0.00* T |
| Supramentale | 11.9 | 1.53 | 0.2 | 12.9 | 1.85 | 0.28 | -1.1 | 7.7 | 0.00* T |
| Pogonion | 9.8 | 2.18 | 0.29 | 11.4 | 2.25 | 0.34 | -1.6 | 14.0 | 0.00* T |
| Menton | 7 | 1.73 | 0.24 | 9 | 2.08 | 0.33 | -2 | 22.2 | 0.00* T |
| Frontal Eminence R | 4.1 | 0.89 | 0.14 | 5 | 1.23 | 0.33 | -0.9 | 18.0 | 0.00* T |
| Frontal Eminence L | 4.2 | 1.11 | 0.18 | 4.9 | 1.2 | 0.32 | -0.7 | 14.2 | 0.04* T |
| Mid-supraorbital R | 6.5 | 1.27 | 0.17 | 8.6 | 1.33 | 0.2 | -2.1 | 24.4 | 0.00* T |
| Mid-supraorbital L | 6.4 | 1.37 | 0.18 | 8.7 | 1.33 | 0.2 | -2.4 | 26.4 | 0.00* T |
| Mid-infraorbital R | 5.3 | 1.47 | 0.2 | 5.8 | 1.51 | 0.23 | -0.5 | 8.6 | 0.07 M |
| Mid-infraorbital L | 5.5 | 1.58 | 0.21 | 5.9 | 1.65 | 0.25 | -0.4 | 6.7 | 0.07 M |
| Malar R | 21.4 | 2.57 | 0.34 | 22.8 | 2.75 | 0.41 | -1.4 | 6.1 | 0.01* T |
| Malar L | 21.7 | 2.38 | 0.32 | 22.9 | 2.8 | 0.42 | -1.2 | 5.2 | 0.02* T |
| Lateral Orbital R | 9.1 | 1.73 | 0.23 | 8.3 | 1.46 | 0.22 | 0.8 | -9.6 | 0.02* M |
| Lateral Orbital L | 9.2 | 1.74 | 0.23 | 8.3 | 1.36 | 0.2 | 0.8 | -10.8 | 0.02* M |
| Zygion R | 7.8 | 1.79 | 0.24 | 9.2 | 1.85 | 0.28 | -1.4 | 15.2 | 0.00* M |
| Zygion L | 7.8 | 1.9 | 0.25 | 9.1 | 1.87 | 0.28 | -1.3 | 14.2 | 0.00* M |
| Supraglenoid R | 10.7 | 1.9 | 0.25 | 12.8 | 1.57 | 0.23 | -2.1 | 16.4 | 0.00* T |
| Supraglenoid L | 10.7 | 1.84 | 0.25 | 12.8 | 1.6 | 0.24 | -2.1 | 16.4 | 0.00* T |
| Gonion R | 12.8 | 3.65 | 0.49 | 18.3 | 5.54 | 0.83 | -5.5 | 30.0 | 0.00* M |
| Gonion L | 13 | 3.88 | 0.52 | 17.8 | 5.49 | 0.82 | -4.8 | 26.9 | 0.00* M |
| Ectomolare$^2$ R | 27 | 3.62 | 0.48 | 30.3 | 3.29 | 0.49 | -3.3 | 10.8 | 0.00* M |
| Ectomolare$^2$ L | 27.2 | 3.64 | 0.49 | 30.2 | 3.44 | 0.51 | -2.9 | 9.9 | 0.00* M |
| Occlusal Line R | 20.3 | 2.82 | 0.38 | 24.6 | 3.17 | 0.47 | -4.4 | 17.4 | 0.00* T |
| Occlusal Line L | 20.4 | 2.86 | 0.38 | 24.3 | 3.21 | 0.48 | -3.8 | 16.0 | 0.00* T |
| Ectomolare$_2$ R | 24.9 | 3.06 | 0.41 | 28.2 | 3.96 | 0.59 | -3.4 | 11.7 | 0.00* T |
| Ectomolare$_2$ L | 24.8 | 3.2 | 0.43 | 28.4 | 3.48 | 0.52 | -3.6 | 12.6 | 0.00* T |

R–right; L–left; T–*t*-test; M–Mann Whitney.

* $P < 0.05$.

**; ***Differences were calculated between female means compared to male means.

Compared to the previous research [18], that produced a FSTT chart from a Brazilian Southeastern sample, it was clear that the heterogeneity of the Brazilian population is marked by a diverse population affinity [2]. Analyzing the average of the differences among the Southeastern and Midwestern samples, the negative variation shows a predominance of deeper FSTTs among the Midwestern sample when compared to the Southeastern sample.

Considering the differences related to sex in the samples of the Southeastern [18] and Midwestern regions (this paper), statistical differentiation was observed. In males, differences were more evident in five landmarks on the midline and eight bilateral points. Quantitatively, that variance did not surpass 2.5 mm, except the ectomolare$_2$ with 3.1 mm. Proportionally, it varied

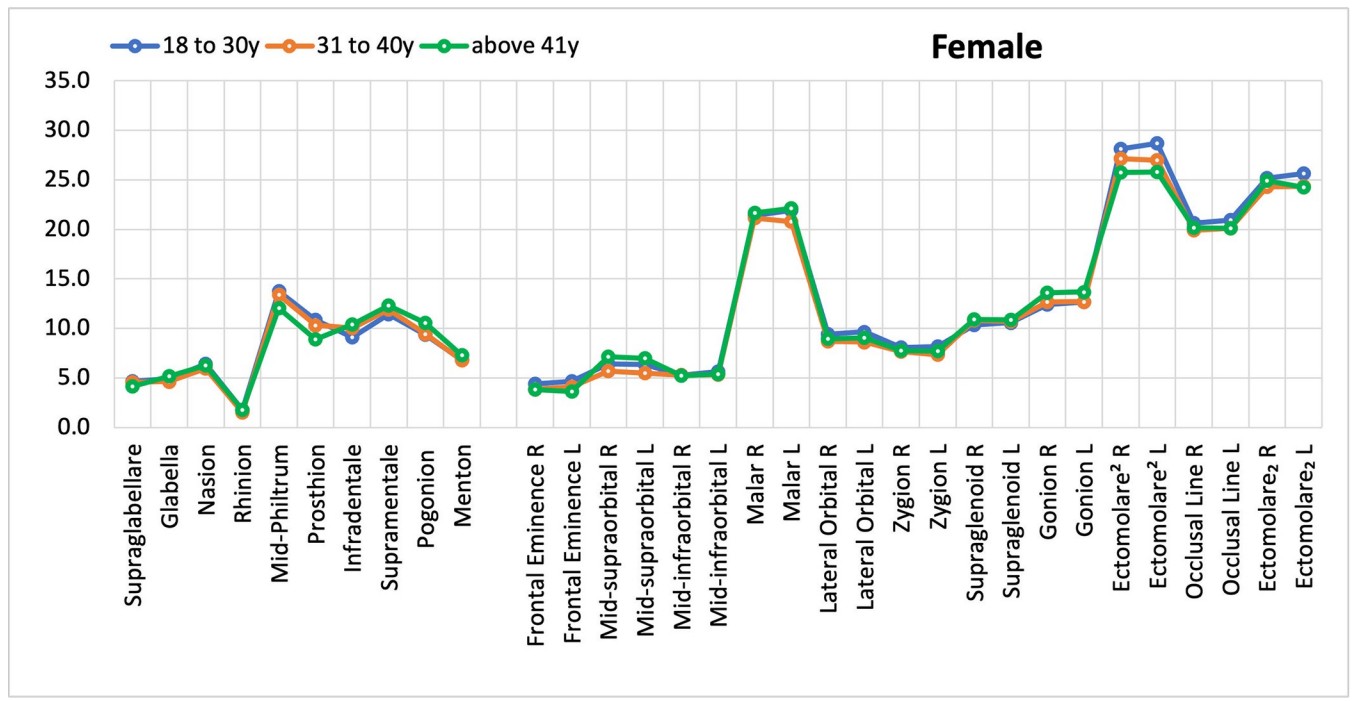

**Fig 5. FSTT means: Comparison among age groups in males.**

between 10 to 15% in most cases with a few landmarks showing higher values. Within females, studies indicate fewer divergent landmarks (three on the midline and eight bilateral) where only malar showed a difference above 2.5 mm (3.0 mm). These results reveal high

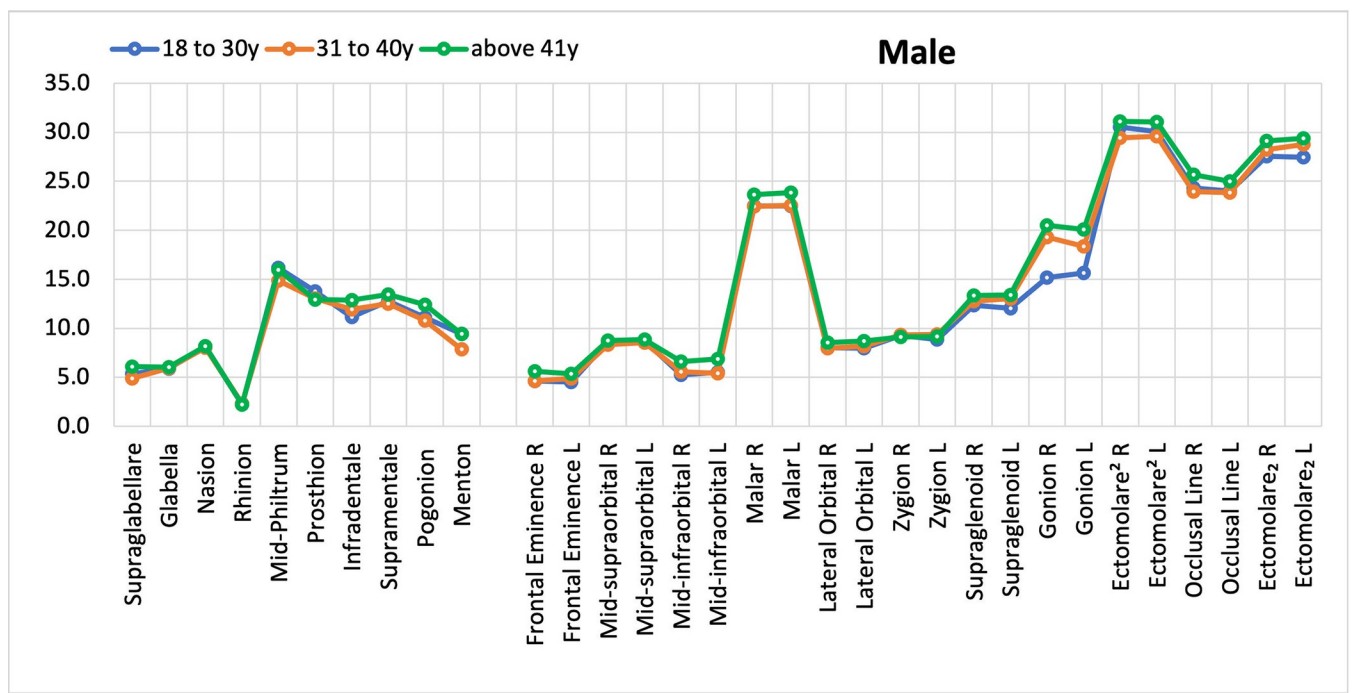

**Fig 6. FSTT means: Comparison among age groups in females.**

**Table 4. FSTT (in millimeters) comparison among SE and MW regions, in both sexes.**

| Landmark | SE | MW | Mean Difference (mm)** | Mean Difference (%)*** | T Test Sig. | SE | MW | Mean Difference (mm)** | Mean Difference (%)*** | T Test Sig. |
|---|---|---|---|---|---|---|---|---|---|---|
| | | | **Female** | | | | | **Male** | | |
| Supraglabellare | 3.4 | 4.5 | -1.1 | -24.4 | 0.00* | 4.3 | 5.5 | -1.2 | -21.8 | 0.00* |
| Glabella | 5 | 4.9 | 0 | 0.0 | 0.94 | 5.8 | 5.9 | -0.2 | -3.4 | 0.45 |
| Nasion | 5.9 | 6.2 | -0.3 | -4.8 | 0.11 | 7.2 | 8.1 | -1 | -12.3 | 0.00* |
| Rhinion | 1.7 | 1.7 | 0 | 0.0 | 0.61 | 1.9 | 2.3 | -0.3 | -13.0 | 0.01* |
| Mid-Philtrum | 12.3 | 13.1 | -0.7 | -5.3 | 0.02* | 15.1 | 15.7 | -0.6 | -3.8 | 0.16 |
| Prosthion | 9.5 | 10 | -0.5 | -5.0 | 0.17 | 12.4 | 13.3 | -0.9 | -6.8 | 0.02* |
| Infradentale | 11.3 | 9.8 | 1.5 | 15.3 | 0.29 | 11.2 | 11.9 | -0.8 | -6.7 | 0.05* |
| Supramentale | 10.8 | 11.9 | -1.1 | -9.2 | 0.00* | 11.4 | 12.9 | -1.5 | -11.6 | 0.00* |
| Pogonion | 9.4 | 9.8 | -0.4 | -4.1 | 0.42 | 10.8 | 11.4 | -0.7 | -6.1 | 0.21 |
| Menton | 6.9 | 7 | -0.1 | -1.4 | 0.87 | 8.6 | 9 | -0.4 | -4.4 | 0.38 |
| Frontal Eminence R | 3.5 | 4.1 | -0.6 | -14.6 | 0.00* | 4.4 | 5 | -0.6 | -12.0 | 0.09 |
| Frontal Eminence L | 3.4 | 4.2 | -0.8 | -19.0 | 0.00* | 4.5 | 4.9 | -0.5 | -10.2 | 0.21 |
| Mid-supraorbital R | 6.2 | 6.5 | -0.3 | -4.6 | 0.22 | 7.3 | 8.6 | -1.3 | -15.1 | 0.00* |
| Mid-supraorbital L | 6.1 | 6.4 | -0.2 | -3.1 | 0.33 | 7.2 | 8.7 | -1.6 | -18.4 | 0.00* |
| Mid-infraorbital R | 5 | 5.3 | -0.3 | -5.7 | 0.32 | 5.4 | 5.8 | -0.4 | -6.9 | 0.16 |
| Mid-infraorbital L | 4.8 | 5.5 | -0.6 | -10.9 | 0.03 | 5.4 | 5.9 | -0.5 | -8.5 | 0.07 |
| Malar R | 19.4 | 21.4 | -2 | -9.3 | 0.00* | 20.3 | 22.8 | -2.5 | -11.0 | 0.00* |
| Malar L | 18.7 | 21.7 | -3 | -13.8 | 0.00* | 20.3 | 22.9 | -2.6 | -11.4 | 0.00* |
| Lateral Orbital R | 9 | 9.1 | -0.1 | -1.1 | 0.76 | 7.5 | 8.3 | -0.8 | -9.6 | 0.01* |
| Lateral Orbital L | 10.4 | 9.2 | 1.2 | 13.0 | 0.38 | 7.3 | 8.3 | -1 | -12.0 | 0.00* |
| Zygion R | 7.4 | 7.8 | -0.4 | -5.1 | 0.27 | 8.1 | 9.2 | -1.1 | -12.0 | 0.01* |
| Zygion L | 7.5 | 7.8 | -0.3 | -3.8 | 0.43 | 7.8 | 9.1 | -1.4 | -15.4 | 0.00* |
| Supraglenoid R | 10 | 10.7 | -0.7 | -6.5 | 0.06 | 11.4 | 12.8 | -1.4 | -10.9 | 0.00* |
| Supraglenoid L | 9.9 | 10.7 | -0.8 | -7.5 | 0.04* | 11.1 | 12.8 | -1.6 | -12.5 | 0.00* |
| Gonion R | 13.2 | 12.8 | 0.3 | 2.3 | 0.66 | 16.9 | 18.3 | -1.4 | -7.7 | 0.21 |
| Gonion L | 13.2 | 13 | 0.2 | 1.5 | 0.73 | 17.2 | 17.8 | -0.6 | -3.4 | 0.55 |
| Ectomolare[2] R | 26 | 27 | -1 | -3.7 | 0.11 | 28.2 | 30.3 | -2.2 | -7.3 | 0.01* |
| Ectomolare[2] L | 26.3 | 27.2 | -0.9 | -3.3 | 0.17 | 28.2 | 30.2 | -1.9 | -6.3 | 0.01* |
| Occlusal Line R | 20.1 | 20.3 | -0.2 | -1.0 | 0.73 | 22.8 | 24.6 | -1.8 | -7.3 | 0.01* |
| Occlusal Line L | 20.4 | 20.4 | 0 | 0.0 | 0.93 | 23 | 24.3 | -1.3 | -5.3 | 0.07 |
| Ectomolare[2] R | 23.5 | 24.9 | -1.3 | -5.2 | 0.03* | 25.1 | 28.2 | -3.1 | -11.0 | 0.00* |
| Ectomolare[2] L | 24 | 24.8 | -0.8 | -3.2 | 0.18 | 25.5 | 28.4 | -2.9 | -10.2 | 0.00* |

*$P < 0.05$.

**; *** Differences calculated between the averages of SE sample compared to MW sample.

compatibility among the samples, considering that errors lower than 2.5 mm, may result in high correspondence on a facial pool [31] and differences between $2.5 \leq 2.9$ mm have minimal practical impact [32].

Whether reconstructions should match local facial traits is not under debate, and the use of FSTT tables of a specific ancestry FSTT charts may provide better FFR. However, local changes should not be exclusively related to FSTT [33]. Eyes, lips, nose, and skin features could also be relevant in the differentiation of people from these two different areas in Brazil.

Investigating further into our results, they support that previous to the FFR, a careful anthropological evaluation of the skull regarding sex, age and ancestry variance remain necessary [6] because it provides important information to the forensic professional. However, the use of FSTT mean values divided by sex, age, and ancestry is no guarantee of better recognition rates, and awareness of the complexity of soft tissue displacement caused by aging [34]. The assessment of the age variant was originally made to reduce sample distribution bias when compared to the Southeastern sample [18] but it presented interesting results. Our results show that attention to this matter should not be neglected on reconstructed faces [27].

The same was planned to verify sex differences because the Southeastern research showed statistically significant changes in STT related to sex. In this research, in agreement with previous studies [18, 35–37], the analysis among sexes revealed males tend to possess larger depths than females, excepting the lateral orbital landmark, in which it has been observed a mean difference of 0.8 mm.

However, such statistical difference may not be the only reason for sexual dimorphism features [9, 36, 38]. Landmarks showed FSTT averages variate thickness within a small quantitative interval (<6% average), leaving sex variance among the individuals to skull formation [38].

The variation between males and females was no greater than 3 mm in most points. Only four landmarks exhibited greater discrepancies and those were located at anatomical regions most influenced by the BMI [37], such as gonion (5.5 mm), ectomare$^2$ (3.3 mm), ectomloare$_2$ (3.6 mm) and occlusal line (4.4 mm). Hence, sexual dimorphism was not clearly evidenced in the variation of depths of soft tissue. Despite the statistical differences, the authors found that male and female individuals appear to share similar averages [38]. The human skull is the third best reference for determining sex, after the pelvic bone and postcranial dimensions [39]. Therefore, the differences verified in this study indicate that sexual dimorphism is probably based, mainly, on the skull morphology.

Multiple facial morphological and physiological alterations act on the ageing process of the face. The initiation and progression vary between individuals, sexes, and populations [40]. In this study, meaningful differences were observed in the FSTT among the age groups, notably in females, at the landmarks located on the medium and inferior face. The FSTT averages of the landmarks mid-philtrum, prosthion and ectomolare$^2$ tend to decrease progressively with age, reproducing the morphological alteration patterns that occur in facial ageing, demonstrating a coherence among the data. With the advancing of age, the average face volume is permanently lost from both from superficial and deep fat tissue [41], and a redistribution of this volume occurs that causes slenderizing of the superior lip more than the inferior [42]. Bone remodeling in that region results in a decrease of the overlying soft tissue support area. This phenomenon, along with the effect of gravity, characterizes the average ageing of the face [34, 43].

## Conclusions

The measurements analyzed in this research, when compared to those collected in the Southeast region of Brazil, are statistically different at many landmarks. However, observing the averages in a specific FSTT chart, indicated that regional differences between the two samples are quantitatively small and they should have minimal influence on facial reconstructions. Therefore, the use of specific FSTT charts is not necessary when performing FFR of individuals from Southeast and Midwest Brazil.

## Supporting information

**S1 Table. ICC intra- and interexaminer.**
(PDF)

**S2 Table. FSTT means in age groups in both sexes.**
(PDF)

## Author Contributions

**Conceptualization:** Deisy Satie Moritsugui, Rodolfo Francisco Haltenhoff Melani.

**Data curation:** Deisy Satie Moritsugui, Thiago Leite Beaini.

**Formal analysis:** Flavia Vanessa Greb Fugiwara, Luiz Eugênio Nigro Mazzilli.

**Investigation:** Flavia Vanessa Greb Fugiwara, Flávia Nicolle Stefani Vassallo.

**Methodology:** Deisy Satie Moritsugui, Thiago Leite Beaini, Rodolfo Francisco Haltenhoff Melani.

**Project administration:** Rodolfo Francisco Haltenhoff Melani.

**Resources:** Deisy Satie Moritsugui, Rodolfo Francisco Haltenhoff Melani.

**Supervision:** Thiago Leite Beaini, Rodolfo Francisco Haltenhoff Melani.

**Validation:** Deisy Satie Moritsugui, Luiz Eugênio Nigro Mazzilli.

**Visualization:** Flavia Vanessa Greb Fugiwara, Flávia Nicolle Stefani Vassallo.

**Writing – original draft:** Deisy Satie Moritsugui, Flávia Nicolle Stefani Vassallo, Thiago Leite Beaini.

**Writing – review & editing:** Deisy Satie Moritsugui, Flavia Vanessa Greb Fugiwara, Luiz Eugênio Nigro Mazzilli, Rodolfo Francisco Haltenhoff Melani.

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
