## [Decision Letter · Decision Letter 0]

23 Feb 2022

PONE-D-22-02392Facial Soft Tissue thickness in Forensic Facial Reconstruction: Impact of regional differences and age effectPLOS ONE

Dear Dr. Moritsugui

Thank you for submitting your manuscript to PLOS ONE. After careful consideration, we feel that it has merit but does not fully meet PLOS ONE’s publication criteria as it currently stands. Therefore, we invite you to submit a revised version of the manuscript that addresses the points raised during the review process.

The reviewers have provided detailed recommendations for revision (see below), which must be addressed in full.

We look forward to receiving your revised manuscript.

Kind regards,

Caroline Wilkinson, PhD

Academic Editor

PLOS ONE

Journal Requirements:

2.In your Data Availability statement, you have not specified where the minimal data set underlying the results described in your manuscript can be found. PLOS defines a study's minimal data set as the underlying data used to reach the conclusions drawn in the manuscript and any additional data required to replicate the reported study findings in their entirety. All PLOS journals require that the minimal data set be made fully available. For more information about our data policy, please see http://journals.plos.org/plosone/s/data-availability.

Reviewers' comments:

Reviewer's Responses to Questions

**Comments to the Author**

1. Is the manuscript technically sound, and do the data support the conclusions?

Reviewer #1: Partly

Reviewer #2: Yes

2. Has the statistical analysis been performed appropriately and rigorously? 

Reviewer #1: Yes

Reviewer #2: Yes

3. Have the authors made all data underlying the findings in their manuscript fully available?

Reviewer #1: Yes

Reviewer #2: Yes

4. Is the manuscript presented in an intelligible fashion and written in standard English?

Reviewer #1: No

Reviewer #2: No

5. Review Comments to the Author

Reviewer #1: Thank you for the opportunity to review this paper.

Language editing needed – I indicated these by yellow highlighting in the attached document as they were too much to correct.

Overall, I think the research is sound, although not very innovative in the sense that it repeats what others have said before. We will need a better discussion of the results, and based on this and other studies clear guidance on the way forward with regard to the use of STT tables. What is the most important – age? Sex? Population?

Below are some comments, which should be read in conjunction with the comments in the text itself (made as sticky notes and highlights in yellow).

Title: Add “in Brazil” at the end

Abstract:

Needs editing as indicated in the attached document. Please provide the sample sizes.

Introduction:

The debate about the use of population-specific STT is not fully explored. This needs more detail.

We need to know more as to why the two populations are expected to be different – i.e., the background of the samples.

The Intro needs to better reflect current debates around the issues raised.

M and M

We need info on the age and sex distribution of the sample (found this later in Results – should be in M and M). Sample sizes in each sex-age cohort quite small (as few as n = 13). This is a shortcoming that should be discussed later on. We also need an indication of the sample sizes of the comparative sample.

I don’t know the software used – cant comment on it

I assume the individuals were in a sitting position when scanned? – this needs better discussion also with regards to best practice and current guidelines.

Inter- and intra-observer repeatability assessment needs better explanation

Comparison between the two regions – are the people from the two regions supposed to be different? Why would we expect them to be different? This needs explanation.

Results

The inter- and intra-observer results are too uninformative, we need the actual results. Inter-observer above 95% is really unusual.

Table with sample sizes should be in the M and M. In the end, the sample size per sex-age group is really small (with as few as 13 individuals)

Differences between the two compared groups should be discussed in more detail – how many measurements differed significantly? It seems there were quite a few. When does one decide that these differences are enough to use population-specific data? How much difference would this make in an actual reconstruction? In the literature it was suggested that one should look not so much at the absolute difference in mm, but rather in % of the actual difference (e.g., a 2 mm difference in a measurement of 10 mm may not make much of a difference in a reconstruction, but 2 mm when the average is 5 mm will be more pronounced). Think it was a paper by Briers? This warrants better Discussion (in the Discussion section).

Discussion

Changes with age cannot be ascribed to bone losses only – the soft tissues as well as the hard tissues undergo changes. Changes with age needs a more thorough discussion with reference to the literature– seems that in this study this was more important than differences between populations from different regions – would you advise that separate tables are used for older and younger groups? But not for different populations?

Miscegenation is a strong word to use, with negative connotations. Please rephrase.

It is stated that the regional differences have an insignificant impact on facial soft tissue thicknesses and consequently on FFR result. This last part – the actual result – has not been tested in this study and this statement can thus not be substantiated. It may be suggested, but was not proven by this study.

Overall, we need to be advised on the way forward, based on the results from this (and other) studies. The last sentence of the conclusion seems to contradict all of what had gone before – do we still need to take population, sex etc into account or not?

Reviewer #2: Overall, the manuscript is written well and logically laid out. The authors have provided details of demonstrated rigour with regards to the experimental design and subsequent analysis through inter and intra observer error, tests for normality, and bootstrapping where appropriate. A narrative is built up throughout the paper by describing the methodology in a chronological fashion, which I found to be well considered. All supporting data appears to have been made fully available.

The consideration of the level of specificity and granularity required with FSTTs is an important one, and it was interesting to see that the differences at regional level for Brazil were insufficient to justify multiple FSTT datasets.

I’ve noted a few suggestions below. Additionally, I have made some suggested edits for the English, as in some cases it was a little unclear what you were intending to say. These are small suggestions for the authors to consider, but I would encourage the inclusion of further clarification around some of the points raised:

General comments:

• You cover the literature in terms of the ‘success’ of population specific datasets in FFR, and that incongruent FSTTs still yield a recognisable face, but that congruent population specific datasets might result in a more ‘accurate’ FFR, although differences in protocols might suggest insufficient evidence for this. I wonder if you could provide some more background on the importance of FSTTs in the recognition process. I.e. FSTTs mainly cover the contours of the face, not the estimation of facial features. Literature suggests facial features and their configuration are more important for familiar face recognition. Furthermore, given the variation of FSTTs in the facial contour due to BMI and ageing, we have a greater tolerance for inaccuracies in these areas. I think it would be useful for you to provide some background on this and to discuss the relative importance of FSTT datasets in the Facial reconstruction and subsequent recognition process. You touch on this briefly in the discussion, but more is needed.

• Lines 273 – 283: Can you provide information of the distribution of samples in the above 41yrs age bracket? i.e. the range? As you mention, age-related changes to the soft tissues change in ‘type’ as age increases. For example, someone in their 40s may have superficial age related changes, caused by a change to the dermis. But those in their 60s and 70s are more likely to also exhibit more extreme volume/morphological changes, such as ptosis etc. which would have a bigger impact on the position of FSTTs. Given this, it would be useful to know the age range for that bracket, and if there was a wide range, why the sample was collapsed into one group. Especially given that a large portion of this manuscript discusses age-related changes.

• Do you anticipate any plans for future research in this area? Given that you observe morphological variation in faces between regions of Brazil, enough to warrant your investigation into FSTTs, might it be worth investigating parameters for feature estimation between these regions? It would be great to see this research carried out.

Formatting edits:

• Perhaps increase the size of the annotation text on figures 2 and 3? If I download the figures I can see it more clearly when zoomed in, but they are barely readable when in article format.

• Ectomolare could be better differentiated rather than using superscript and subscript 2. Whilst it represents upper and lower, I think it’s quite hard to see visually. Maybe 1 and 2 is better?

Grammatical edits:

• Table 1: description for Occlusal line. “where the occlusal line meets de mandible”. EDIT to “where the occlusal line meets the mandible”

• LINE 30: “…reconstruction aims to assemble”

• LINE 31: It’s not just next of kin, it could be friends etc. I would rephrase this

• LINE 38: “…which are reflected in the facial features” (see my third general comment as well)

• LINE 38: “This paper aimed to measure and compare….”

• LINE39:”…to ascertain the need for specific datasets for different regions”

• LINE 44: “As the age of the participants increased…”

• LINE 48: High compatibility was observed when comparing…..

• LINE 51: “Therefore, considering these two geographic regions, the need for applying different datasets has been shown to be unnecessary”

• LINE 58: “Forensic Facial Reconstruction (FFR) has an important role in helping to identify individuals that are unable to be identified by primary methods due to post-mortem changes, or by lack of ante-mortem information.”

• LINE 63: “…a public campaign is issued, aiming to illicit a response from the public as to who the remains may belong to. Following this, a formal identification can then take place”.

• LINE 69: “More recently, computerised face sculpture can be adopted by using digital modelling software, …”

• LINE 75: “In scientific literature….”

• LINE 91: I think “miscegenation” might be considered a derogatory term, and also relates more closely to Race rather than Ancestry. Consider rephrasing/choosing another term. It’s also used again throughout the manuscript.

• LINE 103: “The sample was composed of 101 cone beam…”

• LINE 133: “The Beaini et al……”

• LINE 137: “In total….”

• LINE 148: …CBCT were imported into the software….”

• LINE 153: “Prior to the gathering...”

• LINE 161: “spreadsheet Microsoft Excel for MAC was used….”

• LINE 163: “establishing…..”

• LINE 169: “…., and with non-normal data…”

• LINE 170: “analysis from a one-way ANOVA was applied to investigate any differences among the…..”

• LINES 181-183: “ ..composed of 101 exams, was divided into subgroups for age and sex, shown in Table 3.”

• LINES 205-6: …” that the male sample exhibited higher FSTT values compared to the female sample, except for Orbital Lateral…..”

• LINES 207-8:” The major thickness discrepancies between the sexes were found on….

• LINES 211 – 212: “Regarding age, an ANOVA was used, on both sexes, to compare averages between the age groups…”

• LINE 216: “Figs 4 and 5…”

• LINE 218: “specifically for females…”

• LINE 229: “..a predominance of higher mean FSTTs on the MW sample…..”

• LINE 234: “….using CBCT result in reliable….”

• LINES 236-7: “….showed that the Beaini protocol (17) yields excellent general reproducibility.”

• LINE 245: Consider using “Practitioner” rather than “forensic professional”

• LINES 250-251: “…revealed a tendency for males to exhibit greater tissue depths than females, except the Lateral….”

• LINES 252-255: Consider rephrasing. It’s not overly clear what is meant here?

• LINES 256-257: “…showed slight differences which were less than 3mm. Only four landmarks…..”

• LINE 261: “….variation of the soft tissue depths.”

• LINE 272: “..located in the mid and lower face.” Same for LINE 275

• LINE 309: “…divided into five regions..”

• LINE 312: “… of settlement and continuous migration….”

• LINE 321: “..differences were observed for both sexes. In males, differences were observed for five landmarks on the midline…”

• LINE 330: “…differences between the two studied….”

• LINE 331: “…Nonetheless, the ageing process…”

• LINE 333: “…The establishment of a biological profile…”

• LINE 335: “ …before selecting a FSTT dataset for FFR.”

6. PLOS authors have the option to publish the peer review history of their article (what does this mean?). If published, this will include your full peer review and any attached files.

Reviewer #1: No

Reviewer #2: No

---

## [Author Response · Author response to Decision Letter 0]

29 Apr 2022

The authors would like to thank the reviewers for their valuable comments and excellent suggestions to improve the quality of our manuscript.

The authors have revised all points brought by the reviewers. We sincerely hope that the information presented in this manuscript provides some answers and opens novel questions to advance future research.

---

## [Editor Report · Decision Letter 1]

8 Jun 2022

PONE-D-22-02392R1Facial Soft Tissue thickness in Forensic Facial Reconstruction: Impact of regional differences in BrazilPLOS ONE

Dear Dr. Moritsugui,

Thank you for submitting your manuscript to PLOS ONE. After careful consideration, we feel that it has merit but does not fully meet PLOS ONE’s publication criteria as it currently stands. Therefore, we invite you to submit a revised version of the manuscript that addresses the points raised during the review process.

Please make minor corrections as specified in the attached document.==============================

We look forward to receiving your revised manuscript.

Kind regards,

Caroline Wilkinson, PhD

Academic Editor

PLOS ONE

Journal Requirements:

Additional Editor Comments (if provided):

Further minor changes necessary - these are highlighted in the attached document.
---

## [Author Response · Author response to Decision Letter 1]

10 Jun 2022

The authors would like to thank the reviewers and the Editor for their valuable comments that definitely improved the quality of the manuscript entitled “Facial soft tissue thickness in forensic facial reconstruction: Impact of regional differences in Brazil” (PONE-D-22-02392R1).

Additional Editor Comments:

Further minor changes necessary - these are highlighted in the attached document.

Action: As requested by the Editor, the authors revised the paper and all the necessary changes (highlighted) were made.

---

## [Editor Report · Decision Letter 2]

17 Jun 2022

PONE-D-22-02392R2Facial Soft Tissue thickness in Forensic Facial Reconstruction: Impact of regional differences in BrazilPLOS ONE

Dear Dr. Moritsugui,

Thank you for submitting your manuscript to PLOS ONE. After careful consideration, we feel that it has merit but does not fully meet PLOS ONE’s publication criteria as it currently stands. Therefore, we invite you to submit a revised version of the manuscript that addresses the points raised during the review process.

Please make the one missed correction as the current sentence does not make sense.

We look forward to receiving your revised manuscript.

Kind regards,

Caroline Wilkinson, PhD

Academic Editor

PLOS ONE

Journal Requirements:

Additional Editor Comments (if provided):

All corrections have been made except one:

Line 27 should read: 'appearance of a face over a skull, in order to lead to recognition of that individual, making possible the application of primary identification methods'.
---

## [Author Response · Author response to Decision Letter 2]

17 Jun 2022

The authors would like to apologize for this lapse, and as requested by the Editor, the authors revised the paper and the correction in Line 27 (highlighted) was made.

---

## [Editor Report · Decision Letter 3]

22 Jun 2022

Facial Soft Tissue thickness in Forensic Facial Reconstruction: Impact of regional differences in Brazil

PONE-D-22-02392R3

Dear Dr. Moritsugui,

We’re pleased to inform you that your manuscript has been judged scientifically suitable for publication and will be formally accepted for publication once it meets all outstanding technical requirements.

Kind regards,

Caroline Wilkinson, PhD

Academic Editor

PLOS ONE
---

## [Editor Report · Acceptance letter]

7 Jul 2022

PONE-D-22-02392R3 

Facial soft tissue thickness in forensic facial reconstruction: impact of regional differences in Brazil 

Dear Dr. Moritsugui:

I'm pleased to inform you that your manuscript has been deemed suitable for publication in PLOS ONE. Congratulations! Your manuscript is now with our production department. 

Kind regards, 

on behalf of

Professor Caroline Wilkinson 

Academic Editor

PLOS ONE